# The Efficacy of Honey Compared to Silver Sulfadiazine for Burn Wound Dressing in Superficial and Partial Thickness Burns—A Systematic Review and Meta-Analysis

Samatar Osman [1,2,*], Hamza Umar [2], Yousuf Hashmi [2], Anam Jawaid [2] and Zubair Ahmed [1,2,3,4,*]

1   Institute of Inflammation and Ageing, University of Birmingham, Birmingham B15 2TT, UK
2   College of Medical and Dental Sciences, University of Birmingham, Birmingham B15 2TT, UK
3   Surgical Reconstruction and Microbiology Research Centre, National Institute for Health Research, Queen Elizabeth Hospital, Birmingham B15 2TH, UK
4   Centre for Trauma Sciences Research, University of Birmingham, Birmingham B15 2TT, UK
*   Correspondence: sxo636@student.bham.ac.uk (S.O.); z.ahmed.1@bham.ac.uk (Z.A.)

**Abstract:** Burn dressings play a vital role in protecting the patient from infection and aiding in the wound healing process. At present, the best burn wound dressing remains unknown. This study aimed to assess the efficacy of honey versus silver sulfadiazine dressing (SSD) for the treatment of superficial and partial thickness burns. We performed a systematic review and meta-analysis using the PubMed, MEDLINE and Embase databases to find relevant randomised control trials (RCTs) for inclusion. The outcomes measures included complete burn wound healing time, the proportion of wounds rendered sterile and subjective pain relief associated with the respective dressing type. This review was completed in line with PRISMA guidelines and has been registered with PROSPERO (Study ID: CRD42022337433). All studies in the English language that assessed honey versus SSD for patients with superficial or partial thickness burns were included. Quality and risk of bias assessments were performed using the Cochrane RoB2 tool. Seven studies were identified: totalling a population of 582 patients. From three studies, meta-analysis showed no significant difference in complete wound healing time ($p = 0.06$). Meta-analysis from five studies highlighted an overall significant difference favouring honey dressing in the proportion of wounds rendered sterile at day 7 post-injury (OR 10.80; 95% CI [5.76, 20.26]; $p < 0.00001$; $I^2 = 88\%$). We conclude that honey dressings may be as or more effective than SSD in the treatment of superficial and partial thickness burn injuries. However, due to the low quality of available studies in this field, further research is necessary to establish the optimum burn dressing. Ideally, this should be conducted in the form of prospective three-arm RCTs in accordance with the CONSORT statement.

**Keywords:** honey; silver; burn wounds; wound healing; dressings



## 1. Introduction

Burn injuries consist of a spectrum of traumatic insults to the body and occur when the skin comes into direct contact with a heat source at or above the energy threshold to cause permanent damage [1,2]. Burns can be differentiated by the mechanism of injury; these include thermal (e.g., scald, flame, cold), chemical, electrical, contact, friction, and radiation [3]. Burns injuries must be assessed and classified correctly via the extent and depth of the burn as this ultimately affects outcomes. The British Burn Association (BBA) classifies burns into one of four types: epidermal, superficial partial thickness, deep partial thickness, and full thickness [4].

The health and socioeconomic implications of burns are staggering. From a local level, the skin loses its protective capabilities against pathogens, and the moist wound environment provides a favourable home for bacteria to multiply [5]. Severe burn injuries can also lead to a systemic dampening of the immune response triggered by the injured

tissue which further exacerbates infection risk. This can result in sepsis which accounts for the commonest cause of death in burns patients [6,7]. Infections to the burn wound correlate to increased wound healing time, hospital length of stay and mortality. The impact of burn injuries on the individual includes loss of occupation, physical disfigurement, decreased quality of life, social isolation, and psychological impairment [8–10]. The National Health Service (NHS) estimates that the cost of burn wound care for unhealed wounds is approximately £40,577 per patient over 24 months [11]. Hence, it is paramount that the most efficacious burn wound dressing is made available to all patients to ensure the burn heals as quickly and efficiently as possible.

According to the available literature, the properties of the "ideal burn wound dressing" should include protecting the burn wound from further physical damage and infection, being non-toxic, non-irritant, and alleviating the patient's pain and discomfort [12–14]. Silver sulfadiazine (SSD) is currently considered the gold standard in topical burn treatment as it can be used for both burn infection prophylaxis and treatment [15]. Silver-containing burn dressings release silver into the moist wound environment and its antimicrobial properties are effective against a variety of bacterial, viral and yeast pathogens (although the exact mechanism of action remains unknown) [16]. Although reducing the risk of infection, recent studies have highlighted the adverse effects of SSD in burn treatment which include slowing epithelization, increasing rates of hypertrophic scar formation and development of systemic side effects [17,18].

Honey is a traditional alternative to SSD which is used in the Indian subcontinent. Its high viscosity, high osmolarity, acidic pH and nutrient content promote wound healing and inhibit bacterial growth in the wound site [19]. Furthermore, specific advantages that honey possesses are its effectiveness against antibiotic-resistant strains of bacteria and relatively low cost [20].

The aim of this systematic review was to assess the efficacy of honey versus SSD in superficial and partial thickness burn wound treatment. Primary outcomes included complete wound healing time and proportion of wounds rendered sterile, whilst the secondary outcome measure was reported pain and discomfort of the topical agent.

## 2. Materials and Methods

### 2.1. Literature Search

This systematic review and meta-analysis were conducted in accordance with the Preferred Reporting Items for Systematic Reviews and Meta-Analysis (PRISMA) protocols [21]. A comprehensive systematic literature search of PubMed, Embase and MEDLINE electronic databases was conducted by two independent researchers (S.O., H.U.). Articles were searched across the electronic databases from inception to 6th June 2022. The search terms were kept broad due to the niche topic area; allowing the researchers a wider pool of potential studies to select from. The following Boolean operators were utilised in the database searches: ("Honey" AND "Burn"). The full search strategy with included search terms is available in File S1. The study protocol has been registered with PROSPERO (Study ID: CRD42022337433) to ensure transparency in the review process, help avoid duplication and to reduce the likelihood of reporting bias [22].

### 2.2. Eligibility Criteria

Table 1 illustrates the inclusion and exclusion criteria for our review. Searches on all databases were restricted to the English language, but date restrictions were not applied. Therefore, all studies published from database inception to the date of search (6 June 2022) were eligible for inclusion. Any disagreements with regards to the eligibility criteria were resolved by discussion with the senior author (Z.A.).

### 2.3. Study Selection

Two independent researchers (S.O., H.U.) screened the titles and abstracts of the articles retrieved via the literature search after excluding all duplicated articles. A full-

text screen was conducted of the remaining articles considered for inclusion. Consensus on discrepancies was achieved via discussion amongst the primary researchers, or via consulting the senior researcher (Z.A.) Furthermore, S.O. explored the reference lists of the reviewed articles to identify additional studies.

**Table 1.** Eligibility Criteria.

| PICOS | Inclusion Criteria | Exclusion Criteria |
| --- | --- | --- |
| Population | Adult and paediatric patients with superficial or partial thickness burns | Patients with full thickness burns |
| Intervention | Honey dressing | N/A |
| Comparison | SSD cream or dressing | Other treatment modalities (e.g., early excision and grafting) |
| Outcome Measures | Primary outcomes: complete wound healing time, proportion of wounds rendered sterile, Secondary outcome: subjective pain relief | N/A |
| Study Design | Randomised controlled trials | Observational studies, animal studies, reviews, abstracts, case reports or quality improvement projects |

### 2.4. Data Extraction

A data extraction Microsoft Excel spreadsheet was completed by two authors (S.O. and Y.H.) and any disagreements were resolved by collaboration with the senior author (Z.A). For each included study, the following data were extracted: study location, demographic data, intervention and comparator information, duration of the study and the reported outcomes.

The primary outcome measure extracted was mean burn wound healing time and proportion of burn wounds rendered sterile. This is because these are objective measures of assessing the effectiveness of honey versus SSD as a burn wound dressing and are also the primary outcomes most frequently utilised in randomised control trials (RCTs). The secondary outcome measure was subjective pain relief due to the known anti-inflammatory properties of honey.

### 2.5. Risk of Bias

The Cochrane RoB2 tool was used to undertake a risk of bias assessment in the RCTs using the templates provided by the Cochrane Group [23]. Two researchers (H.U., A.J.) completed the template assessing the risk of bias over the five domains: D1, Risk of bias arising from the randomization process; D2, Risk of bias due to deviations from the intended interventions; D3, Missing outcome data; D4, Risk of bias in measurement of the outcome; and D5, Risk of bias in the selection of the reported result. For the different domains, a score of low, moderate, or high risk of bias was given. Following this, an overall score was applied to each article included in this study. The overall risk of bias was judged by each individual researcher and any discrepancies were resolved by discussion with the senior author (Z.A.). Data were extracted into Microsoft Excel and a summary diagram and risk of bias in individual studies were compiled.

### 2.6. Statistical Analysis

Assessment of heterogeneity was done by examining the differences across studies for methodological heterogeneity. We used Review Manager (RevMan 5.3, Cochrane Informatics & Technology, London, UK) to determine the Q and $I^2$ statistics (in percentages) to establish variation between the studies attributed to heterogeneity. A meta-analysis was conducted in RevMan 5.3 (Cochrane Informatics & Technology, London, UK), using the dichotomous data function employing a random effects model.

## 3. Results

### 3.1. Study Selection

The initial search from three databases yielded a total of 422 results, with a further four studies identified from additional records. Following the removal of duplicates, 319 studies were extracted and imported into the Covidence database for screening. At this stage, 302 studies were excluded based on the inclusion and exclusion criteria, resulting in 17 for full-text assessment. A further 10 studies were then excluded following full-text analysis and hence seven RCTs were included in this systematic review. File S2 provides a list of full-text exclusions (*n* = 10). The full PRISMA flowchart is provided in Figure 1.

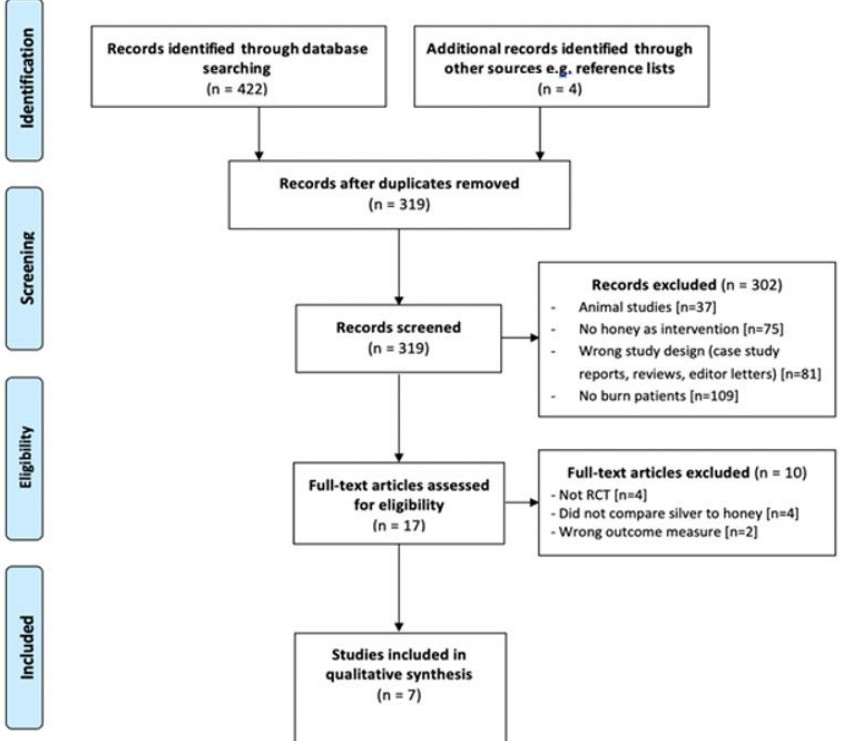

**Figure 1.** PRISMA flow diagram.

### 3.2. Study Characteristics

The studies included in this systematic review were published between April 1991 and January 2011 and were conducted in only two countries: India and Pakistan. All seven studies [24–30] were of an RCT design, with a total of 582 patients randomised to either the honey or SSD group. There was significant variation in patient population characteristics such as age; some studies included both adult and paediatric patients whilst others only had adult patients. Furthermore, the type and extent of burn injuries between studies differed. However, all studies only included patients with superficial and/or partial thickness thermal burns and all the patients had burns of less than 50% total body surface area (TBSA).

Unprocessed, undiluted honey was applied as the intervention in all the RCTs except for one [25] which instead used a commercially available form. All trials utilised honey with sterile dressing as the intervention. SSD cream covered with a sterile dressing was the comparator in many of the studies, but three studies utilised SSD-containing (impregnated) dressings instead [27,29,30]. In addition, there was a difference in the frequency of replacement of wound dressings and duration of patient follow-up between the trials (Table 2). Variation between the study intervention groups included replacing honey dressings twice daily, once daily, or on alternate days.

**Table 2.** Study Characteristics.

| Study | Location | Patient Population | Mean Age in Years (SD) | Intervention (No. of Subjects) | Control (No. of Subjects) | Duration of Study | Assessment of Complete Wound Healing and Sterility | Outcome Measures |
|---|---|---|---|---|---|---|---|---|
| Baghel et al. 2009 [24] | India | 78 patients with first and second degree burns of less than 50% TBSA | I: 34.5 C: 28.5 | Pure, unprocessed, undiluted honey dressing with sterile gauze everyday (*n* = 37) | SSD Cream with sterile gauze everyday (*n* = 41) | 2 months | Clinical observation and wound swab cultures | Wound healing time, time taken to render wounds sterile, proportion of wounds healed |
| Malik et al. 2010 [25] | Pakistan | 150 patients with partial thickness burns less than 40% on two contralateral body sites (e.g., right and left hand) | 28 (15.94) | Langnese honey twice daily (*n* = 150) | SSD Cream with sterile gauze everyday (*n* = 150) | Until burn wound fully healed | Clinical observation and wound swab culture | Wound healing time, proportion of wounds healed |
| Mashhood et al. 2006 [26] | Pakistan | 50 patients with superficial and partial thickness burns less than or equal to 15% TBSA | 27.4 | Pure, unprocessed, undiluted honey which was applied once daily with a sterile gauze (*n* = 25) | 1% SSD once daily (*n* = 25) | Until burn would healed | Clinical observation and wound swabs for bacterial density and cultures | Wound healing time, time taken to render wounds sterile, pain |
| Subrahmanyam et al. 2001 [27] | India | 100 patients with superficial thickness burns | I: 26.5 (1) C: 25.2 (2) | Undiluted, unprocessed honey dressing replaced once every 2 days (*n* = 50) | SSD-impregnated gauze dressing replaced every 2 days (*n* = 50) | Until burn wound healed | Clinical observation and wound swabs for bacterial cultures and sensitivity determinations | Wound healing time, proportion of wounds rendered sterile, subjective relief of pain |
| Sami et al. 2011 [28] | Pakistan | 50 patients with partial thickness thermal burns involving 5–40% TBSA | N/A, range = 1.5–50 | Pure unprocessed undiluted honey applied once daily (*n* = 25) | 1% SSD cream once daily (*n* = 25) | 2 months | Clinical observation of degree of epithelisation and wound swabs | Wound healing time, number of wounds rendered sterile, pain relief, cost of treatment per % burn |
| Subrahmanyam et al. 1998 [29] | India | 50 patients with superficial thermal burns involving less than 40% TBSA | I: 25.2 C: 26.4 | Pure unprocessed undiluted honey dressing replaced on alternate days (*n* = 25) | SSD gauze replaced daily (*n* = 25) | 1 month | Clinical inspection, biopsies for histological studies and wound swabs | Wound healing time, proportion of wounds rendered sterile |
| Subrahmanyam et al. 1991 [30] | India | 104 patients with superficial thermal burns less than 40% TBSA | N/A, range = 1–65 | Pure unprocessed undiluted honey dressing applied once daily (*n* = 52) | SSD gauze replaced daily (*n* = 52) | 2 months | Clinical observation and wound swabs for culture and sensitivity determinations | Wound healing time, proportion of wounds rendered sterile |

N/A = not available; SSD = silver sulfadiazine; TBSA = total body surface area.

Five of the studies [24,27–30] reported our primary outcome measure of complete wound healing time and/or proportion of wounds rendered sterile (negative swab culture at day seven). Three studies [26–28] reported our secondary outcome measure of subjective pain relief between the honey and SSD groups. Table 2 illustrates an overview of the study characteristics with their respective outcome measures assessed.

### 3.3. Results of Individual Studies—Primary Outcomes

All four studies assessing our primary outcome of complete burn wound healing time reported a statistically significant shorter complete wound healing time. For example,

the average wound healing time in the honey group was 12 days, whereas in the group treated with SSD wound dressing the average was 19 days ($p < 0.001$). Table 3 illustrates the individual study results for this parameter.

**Table 3.** Complete Wound Healing Time Results.

| Study | Honey Group Participants | SSD Group Participants | Honey Healing Time (Days ± SD) | SSD Healing Time (Days ± SD) | Mean Difference | *p*-Value |
|---|---|---|---|---|---|---|
| Baghel et al. 2009 [24] | *n* = 37 | *n* = 41 | 18.1 ± NA | 32.6 ± N/A | −14.5 | <0.05 |
| Subrahmanyam et al. 2001 [27] | *n* = 50 | *n* = 50 | 15.4 (3.2) | 17.2 (4.3) | −1.8 | <0.001 |
| Subrahmanyam et al. 1998 [29] | *n* = 25 | *n* = 25 | 4.92 (3.61) | 8.22 (8.31) | −3.3 | <0.001 |
| Subrahmanyam et al. 1991 [30] | *n* = 52 | *n* = 52 | 9.4 (2.3) | 17.2 (3.2) | −7.8 | <0.001 |

N/A = not available; SD = standard deviation.

Despite the limited number of studies, a meta-analysis was conducted which demonstrated there was a weighted mean difference in complete burn wound healing time between the honey and SSD wound dressing groups of −4.37 days (95% CI [−0.19, 8.94]; $p = 0.06$; $I^2 = 95\%$), but this result was not statistically significant (Figure 2). Therefore, it cannot be concluded with any reasonable certainty that this result is not due to chance.

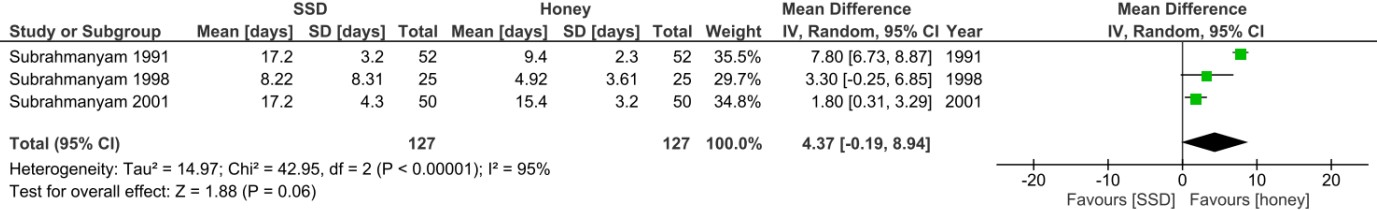

**Figure 2.** Forest plot of mean difference in complete burn wound healing time between honey and SSD burn dressing groups. Green squares = weighted mean difference. Black diamond = overall weighted mean difference.

Furthermore, our additional primary outcome measure was the proportion of burn wounds rendered sterile 7 days after initiation of the wound dressing. A meta-analysis (Figure 3) was conducted for this outcome measure which showed a weighted statistically significant effect favouring the honey group (OR: 10.80, 95% CI [5.76, 20.26]; $p < 0.00001$; $I^2 = 88\%$).

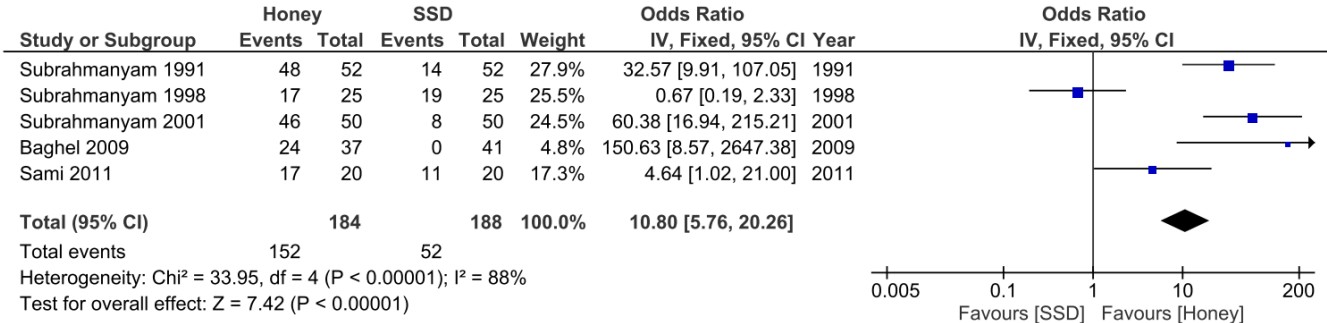

**Figure 3.** Forest plot of the proportion of wounds rendered sterile at day 7 between honey and SSD burn dressing groups. Green squares = weighted mean difference. Black diamond = overall weighted mean difference.

*3.4. Results of Individual Studies—Secondary Outcome*

Although three studies reported pain relief associated with the type of burn wound dressing, only two of these studies provided numerical data. Mashhood et al. [26] evaluated pain relief between the honey and SSD groups. Although no information was provided regarding if the results were statistically significant, the research team found that all patients treated with honey dressing were relieved of pain after 3 weeks, whilst it took 4 weeks for the SSD group to achieve the same outcome.

The study conducted by Sami et al. [28] also provided evidence that honey dressing was more effective for pain relief than SSD. The honey group was associated with statistically significant earlier pain relief; 36% in this cohort were pain-free at day 5 versus 4% in the SSD group ($p = 0.01$). The researchers noted that the mean number of days to achieve pain relief was 12 days in the honey group and 16.8 days in the SSD group.

*3.5. Risk of Bias in Studies*

All seven studies were assessed across five domains, using the RoB-2 tool, to evaluate the potential for risk of bias in methodology and outcomes. Individual study scores are shown in Figure 4.

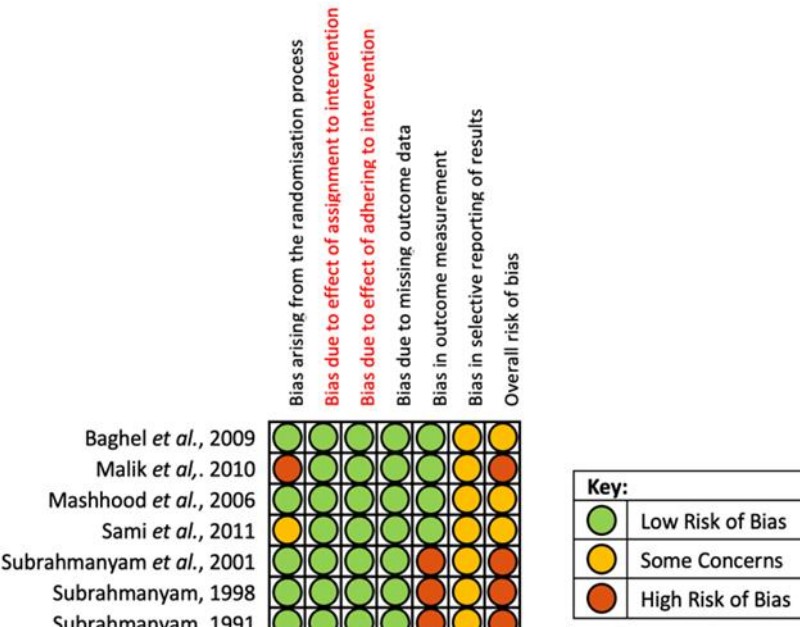

**Figure 4.** Risk of bias assessment summary according to the Cochrane RoB-2 tool [24–30].

Overall, three studies were found to have some concerns, whilst four studies were evaluated to have a high risk of bias. Regarding bias due to randomisation, Sami et al. [28] had some concerns due to utilising non-probability consecutive sampling whilst Malik et al. [25] had a high risk of bias due to inadequate allocation concealment. Only Malik et al. explicitly described the blinding of patients and nursing staff; the increased risk of performance bias in other studies was mitigated by their use of objective primary outcome measures. However, the three studies by Subrahmanyan et al. [27,29,30] assessed the primary outcome via an inherently subjective interpretation of 'complete healing'; resulting in an assessment of high risk of bias for outcome measurement. All studies failed to include an appropriate pre-specified plan for data analysis and reporting. Therefore, it was not possible to determine the level of reporting bias.

**4. Discussion**

Topical application of honey is an ancient remedy used in many low- and middle-income countries (LMICs) for the treatment of various medical ailments including burn

injuries, infected wounds, ulcers and eczema [31,32]. The use of honey offers a cheaper, non-toxic, and non-allergenic alternative to modern burn treatment options such as topical or systemic antimicrobial therapy, surgical debridement of necrotic tissue and more expensive forms of wound dressings [33]. The theory underlying its mechanism of action is comprehensive; in summary, honey has antimicrobial, anti-inflammatory, anti-oedematous and anti-exudative properties [34,35]. In particular, hydrogen peroxide produced in honey has a vital role in stimulating cell proliferation, growth of fibroblasts and regeneration of new capillaries during the wound healing process [36].

This systematic review aimed to ascertain whether honey was more effective than SSD as a wound dressing material in patients with superficial and partial thickness burn injuries. In total from all seven studies, there were 582 patients included in this review. The results of our primary outcome analysis have indicated that honey is more effective than SSD in rendering infected burn wounds sterile. However, from the primary outcome analysis, we cannot conclude that honey is more effective than SSD in reducing the time taken for complete burn wound healing. Only four studies in this systematic review provided data on complete wound healing time, but only three of these were included in the meta-analysis (Baghel et al. did not provide sufficient numerical data) [27,29,30]. Although a difference was noted, it was not significant enough ($p = 0.06$) to meet the threshold required.

Furthermore, a major barrier to interpreting results from this primary outcome analysis and ultimately being able to influence current clinical practice is the high heterogeneity amongst the included studies. This is due to the substantial variation between studies such as participant characteristics, follow-up duration, outcomes reported and more. For our secondary outcome of pain relief, all the studies which included this parameter in their outcomes suggested that honey was more effective than SSD. Nevertheless, it is important to note that pain is a subjective measure, and it is not plausible to correlate the sole introduction of either honey or SSD as a determinant in pain relief. In addition, patients are likely to be on a pain-relief regimen during their treatment in hospital which may differ from patient to patient and also between burn treatment centres which will ultimately affect any conclusions which can be drawn.

Animal studies have consistently demonstrated the effectiveness of honey as an alternative remedy for the management of superficial and partial thickness burns; they have the added benefit of removing the placebo effect found in human clinical trials [37–39]. These studies have demonstrated that the most effective regimen is using unprocessed, floral honey. Furthermore, they have also described a synergistic effect of utilising oral administration of honey alongside topical treatment to maximise effective wound epithelisation [39,40]. For future clinical research in humans, this provides a potential avenue to explore as a burns treatment modality.

The best topical dressing for superficial and partial thickness burns still remains unclear. When applying our findings to clinical practice, there is a clear contradiction as to why SSD is still widely used as a topical burn treatment option when this systematic review found that honey was more effective in sterilising infected wounds, and no significant difference was established with regards to complete wound healing time. Previous research has also shown that SSD can come with harmful side effects and slower reepithelisation [17,18]. Reepithelisation which takes greater than 3-4 weeks is a significant risk factor for the development of hypertrophic scars [41]. Additional benefits of faster burn wound healing include decreased risk of joint contractures, stiffness and ultimately quicker rehabilitation for the patient [42,43]. Delayed wound healing may require surgical intervention in the form of excision and skin graft. However, for patients living in LMICs, financial barriers to treatment are common and with a lack of well-trained and equipped burn surgeons in rural areas, surgery can often lead to more harm than benefit [44,45].

Burn services in LMICs require compromise to deliver the most cost-effective treatment for patients. Hence topical antimicrobial treatments may either be expensive or unavailable for patients. Therefore, these hospitals require a wound dressing, which is effective, simple to apply, economical for the patients and hospital, and with a long shelf life [46]. This is one

factor that makes plant-based traditional remedies popular in these countries. Apart from honey, alternatives that are being used include boiled potato peel dressings (BPPD), and papaya paste (particularly in West Africa). A histological and bacteriological comparative study conducted by Keswani et al. demonstrated that the application of BPPD was shown to significantly reduce or eliminate wound desiccation, increase survival of superficial skin cells and accelerate epithelial regeneration [47]. Although not much clinical evidence is available supporting the use of papaya as a burn wound dressing, Starley et al. suggest that the activity of proteolytic enzymes chymopapain and papain may be effective in desloughing necrotic tissue and preventing burn wound infections [48].

Finally, further research in the form of a definitive, high-quality prospective three-arm RCT is required to compare honey with SSD and either another conventional wound dressing (e.g., hydrocolloid, petroleum impregnated gauze, biosynthetic dressings) or a traditional remedy (BPPD, papaya or banana leaves). The trial should focus on a specific burn pattern with clear inclusion criteria such as patients with partial-thickness burns ≤40% TBS. Patients should be properly randomised whilst concealed allocation and blinded outcome strategies should be applied as best as possible. Furthermore, future trials should include a health-related quality of life measurement to assess the generic and wound-specific impact of the dressing and if possible, cost-effectiveness analysis should also be completed. A recurring theme in the studies included in this systematic review is the lack of consensus on secondary outcomes. The team conducting this systematic review recommends the following outcome measures to be utilised: number of dressing changes, level of pain associated with application and removal, quality of life, hospital length of stay, need for surgery and adverse events.

## 5. Limitations

The limitation in the number of published trials available on databases was a significant hindrance in formulating this systematic review; with many of the included studies being of low-quality evidence not in accordance with the Consolidated Standards of Reporting of Trials (CONSORT) [49]. This meant that certain studies had inadequate reporting of numerical data and therefore were not included in the subsequent meta-analysis despite the best efforts in contacting the relevant authors [25,26]. In addition, articles not written in the English language were excluded from this review which meant some studies may have been missed out during our literature search.

There were further limitations in the designs of the randomised trials. Some of the studies were open trials which meant that there was no concealment for both the researchers or patients in the randomisation process and for which patients received the intervention or control. Subsequently, this is likely to impact the effect the of response to treatment due to performance bias. This review is also disproportionately reliant on studies from one research team in India from one burns unit [27,29,30]. Consequently, this makes it difficult to extrapolate our findings from this systematic review globally due to variations in treatment centres, protocols and microbiological environments.

Another significant limitation of this review is the lack of clear and consistent outcome measures used to establish the efficacy of honey dressing versus SSD. Our review found that honey was more effective at rendering burn wounds sterile compared to SSD on day 7, but the clinical importance of negative wound swabs on day 7 is unclear in the literature. Additionally, the definition of complete wound healing was not explicitly described in all of the studies; and neither was information given as to how it was assessed. Most studies described that clinical observation was undertaken to determine wound healing, but little description was offered beyond this. Baghel et al. stated that if the scar was soft, hypertrophic and/or contracture formation, this was deemed to be incomplete wound healing [24]. In addition, Malik et al. was the only study that identified the healthcare personnel responsible for determining wound healing (a burns surgeon) [25]. Apart from clinical observation, Subrahmanyam et al. was the only study to include histological assessment of reparative activity via biopsy of the wound site [29]. Future trials should be

sure to implement valid and reliable measures of wound healing and infection that can be universally applied in most healthcare settings globally.

**6. Conclusions**

Our findings suggest that honey dressings may be as or more effective than SSD for the treatment of superficial and partial-thickness burn wounds. However, this conclusion is drawn from a small pool of low-quality studies from one area of the world with high heterogeneity, and therefore caution must be exercised in applying honey in routine clinical practice. In addition, the lack of universally agreed primary and secondary clinical outcome measures made a comparison between the different studies challenging. Future studies must ensure they comply with the CONSORT statement to improve the quality of trials. Finally, our study raises question marks as to why SSD remains a benchmark in burn wound dressing in many countries given the evidence from this systematic review and previous studies highlighting that it delays wound reepithelisation.

**Supplementary Materials:** The following supporting information can be downloaded at: https://www.mdpi.com/article/10.3390/traumacare2040043/s1, File S1: full search strategy, File S2: a list of full-text exclusions.

**Author Contributions:** Conceptualization, S.O. and Z.A.; methodology, S.O., H.U., Y.H., A.J. and Z.A.; formal analysis, S.O., H.U., Y.H., A.J. and Z.A.; investigation, S.O., H.U., Y.H. and A.J.; data curation, S.O., H.U., Y.H. and A.J.; writing—original draft preparation, S.O., H.U. and Y.H.; writing—review and editing, S.O., H.U., Y.H., A.J. and Z.A.; supervision, Z.A.; project administration, S.O. All authors have read and agreed to the published version of the manuscript.

**Funding:** This research received no external funding.

**Institutional Review Board Statement:** Ethical review and approval were waived for this study since it is a systematic review of published literature.

**Informed Consent Statement:** Patient consent was waived because no patients or members of the public were involved in the design, conduct of this study, or reporting of this research.

**Data Availability Statement:** All data generated as part of this study are included in the article.

**Conflicts of Interest:** The authors declare no conflict of interest.

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
