# Peer review of "The Efficacy of Honey Compared to Silver Sulfadiazine for Burn Wound Dressing in Superficial and Partial Thickness Burns—A Systematic Review and Meta-Analysis"

_traumacare, doi:10.3390/traumacare2040043_

Round 1

Reviewer 1 Report

This is a very well done systematic review of the literature, I only have limited suggestions for improvement.

Though these elements are present in the body of the article, the abstract may be improved by including elements from the PRISMA guidelines for abstracts, namely the inclusion and exclusion criteria, the methods used to assess bias, the source of funding and the registration of the study.

The determination as to whether a wound has healed can be somewhat subjective – could the authors include in the results section the ways in which wound closure/healing was assessed in each of the studies (perhaps in Table 2?) and perhaps expand slightly on the comment in the limitations section regarding the same? Similarly, could the authors include how it was assessed that a wound was rendered sterile in each study?

Otherwise a strong study and a good addition to the literature.

Author Response

Dear Reviewer,

Thank you for giving me the opportunity to submit a revised draft of my manuscript titled “The Efficacy of Honey compared to Silver Sulfadiazine for Burn Wound Dressing in Superficial and Partial Thickness Burns—A Systematic Review and Meta-Analysis” to Trauma Care. We appreciate the time and effort that you have dedicated to providing your valuable feedback on my manuscript.

We are grateful to the reviewers for their insightful comments on my paper. We have been able to incorporate changes to reflect most of the suggestions provided by the reviewers and have highlighted the changes within the revised manuscript. Here is a point-by-point response to the reviewer’s comments.

Comments from Reviewer 1

  • Comment 1: This is a very well-done systematic review of the literature, I only have limited suggestions for improvement. Though these elements are present in the body of the article, the abstract may be improved by including elements from the PRISMA guidelines for abstracts, namely the inclusion and exclusion criteria, the methods used to assess bias, the source of funding and the registration of the study.

Response: We greatly appreciate the reviewer for these comments and thank them for their kind words. These recommended changes have been made to the abstract. The source of funding has not been added to the abstract due to the word count limit, and because we have explicitly mentioned it in page 12 of the manuscript that this research has received no external funding.

  • Comment 2: The determination as to whether a wound has healed can be somewhat subjective – could the authors include in the results section the ways in which wound closure/healing was assessed in each of the studies (perhaps in Table 2?) and perhaps expand slightly on the comment in the limitations section regarding the same? Similarly, could the authors include how it was assessed that a wound was rendered sterile in each study?

Response: We thank the reviewer for this suggestion of improvement. Table 2 has been expanded to include how wound healing and sterility was assessed by the individual RCTs, although most studies only provided very vague statements with regards to this. Likewise, we have expanded the limitations section to account for these changes.

  • Comment 3: Otherwise a strong study and a good addition to the literature

Response: We greatly appreciate the kind words. The research team unequivocally agrees this systematic review would be a strong addition in the field of burn wound care and the Trauma Care journal.

In addition to the above comments, all spelling and grammatical errors pointed out by the reviewers have been corrected. We look forward to hearing from you in due time regarding our submission and to respond to any further questions and comments you may have.

Sincerely,

Samatar Osman

5th year Intercalating Medical Student (MSc Trauma Science)

University of Birmingham

Reviewer 2 Report

Line 69-71: This statement can be replaced in the discussion section. After analyzing certain literature data, a proof is needed to show the superiority of honey application over SSD. If such evidence already exists, why do the authors focus on this comparison study?

Line 137-138: The relevant expression is not sufficiently understood. (A meta-analysis patient was conducted)

Page 4: Figure 1. It looks that a large number of studies was excluded (n = 302/319). What is the main reason of huge exclusion?

Line 151: why do you mention the studies from only 2 countries? The studies included to this study quite old (before 2011). Is there any other studies from 2011 until now?

There is a previous systematic review which focus on same topic presents the similar outcome with yours.  And also, there is no new RCTs included to your review after 2017. Would you please explain main difference of this review when compared to the study Aziz et al.?

Aziz, Zoriah, and Bassam Abdul Rasool Hassan. "The effects of honey compared to silver sulfadiazine for the treatment of burns: A systematic review of randomized controlled trials." Burns 43.1 (2017): 50-57.

Line 165: What is the application frequency for honey? Did all studies report the dressing time per day? What is the duration between two dressing for both group?

Did dressing application once or twice a day affect the result of the treatment?

In table 3: In table 3, Subrahmanyan et al. reported 3 similar studies and obtained different results. These different results are thought-provoking.

I have not come across a new statement that contributes to the literature. I thank the authors for the fluency of their academic lectures. 

Author Response

Dear Reviewer,

Thank you for giving me the opportunity to submit a revised draft of my manuscript titled “The Efficacy of Honey compared to Silver Sulfadiazine for Burn Wound Dressing in Superficial and Partial Thickness Burns—A Systematic Review and Meta-Analysis” to Trauma Care. We appreciate the time and effort that you have dedicated to providing your valuable feedback on my manuscript.

We are grateful to the reviewers for their insightful comments on my paper. We have been able to incorporate changes to reflect most of the suggestions provided by the reviewers and have highlighted the changes within the revised manuscript. Here is a point-by-point response to the reviewer’s comments.

Comments from Reviewer 2

  • Comment 1: Line 69-71: This statement can be replaced in the discussion section. After analyzing certain literature data, a proof is needed to show the superiority of honey application over SSD. If such evidence already exists, why do the authors focus on this comparison study?

Response: We thank the reviewer for highlighting this mistake. As we already mentioned this point in the discussion section, we have removed lines 69-71 from the revised manuscript due to duplication error.

  • Comment 2: Line 137-138: The relevant expression is not sufficiently understood. (A meta-analysis patient was conducted)

Response: Thank you for pointing this out. This was a mistake; we were meant to write “a meta-analysis was conducted…”. The sentence has been edited to reflect the reviewer’s feedback in the revised manuscript.

  • Comment 3: Page 4: Figure 1. It looks that a large number of studies was excluded (n = 302/319). What is the main reason of huge exclusion?

Response: We thank the reviewer for noticing our lack of clarity. The most common reasons for exclusion during the initial screening stage of titles and abstracts was that many of the papers were of the wrong study design (e.g. case study reports, reviews, etc), no burn patients included as the study population or honey was not utilised as the intervention. We have now updated the PRISMA flow diagram (Figure 1) to illustrate this.

  • Comment 4: Line 151: why do you mention the studies from only 2 countries? The studies included to this study quite old (before 2011). Is there any other studies from 2011 until now?

Response: As this was the study characteristics section, we described that the seven studies we systematically identified in our review were only from India and Pakistan. We acknowledge the reviewer’s comment that most of the studies date pre-2011, however, the study by Sami et al. is from 2011. Our search did not yield any further studies past this date unfortunately. During our discussion section, we urged for more high-quality studies to be published in this field due to the limited pool of resources available. 

  • Comment 5: There is a previous systematic review which focus on same topic presents the similar outcome with yours. And also, there is no new RCTs included to your review after 2017. Would you please explain main difference of this review when compared to the study Aziz et al.? Aziz, Zoriah, and Bassam Abdul Rasool Hassan. "The effects of honey compared to silver sulfadiazine for the treatment of burns: A systematic review of randomized controlled trials." Burns 43.1 (2017): 50-57.

Response: We thank the reviewer for this comment. Although there may be some similarities with the review conducted by Aziz et al., there are significant differences. Firstly, for our primary outcome measure of complete wound healing time, we found no statistically significant difference between the honey and SSD groups, however, Aziz et al. noted a different result of statistical significance in favour of the honey dressing cohort.

Furthermore, the other research team did not include the study conducted by Malik et al. (2010) in their review. Our risk of bias assessment also differed; we concluded moderate-high risk of bias in the RCTs overall, whilst Aziz et al. concluded low-moderate risk.

Additionally, our research team formulated clear, specific areas of future research in the field of burn wound care (mentioned in both the discussion and conclusion sections) including the conduction of a multi-centre, three-arm high quality RCT. We also explored in depth other traditional methods of burn dressings for future research (boiled potato peel dressings and papaya paste) in the discussion section.

We strongly and respectfully believe our systematic review is an important addition to the literature of burn wound care.

  • Comment 6: Line 165: What is the application frequency for honey? Did all studies report the dressing time per day? What is the duration between two dressing for both group? Did dressing application once or twice a day affect the result of the treatment?

Response: The application frequency is described in the study characteristics Table 2, although we have edited this section of the manuscript to provide more clarity. The studies in this review did not explicitly state the exact hourly difference between dressing changes in both groups. However, if changed once a day, it can be assumed the dressings were changed every 24 hours.

The effect of frequency of changing honey dressings is an interesting point. Subrahmanyam et al. 2001 and Subrahmanyam et al. 1998 replaced honey dressings once every 2 days, whilst Subrahmanyam et al. 1991 replaced dressings daily. Table 3 illustrates the differing results which shows that Subrahmanyam et al. 1991 had a greater mean difference in complete wound healing time versus the other two studies. This might support the hypothesis that increased frequency of honey dressing changes may be beneficial, although as stated in our discussion section more high-quality research is needed.

  • Comment 7: In table 3: In table 3, Subrahmanyam et al. reported 3 similar studies and obtained different results. These different results are thought-provoking.

Response: We agree that this is an interesting finding. In the previous comment response, we allude that this may be due to the frequency of dressing changes.

  • Comment 8: I have not come across a new statement that contributes to the literature. I thank the authors for the fluency of their academic lectures.

Response: We respectfully disagree with this statement as mentioned earlier. Our findings raise important concerns as to why silver sulfadiazine is still being used today as a mode of burn wound dressings. This systematic review also changes our perception of the use of traditional remedies for the treatment of patients with thermal injuries, which have been used for centuries in the Indian subcontinent.

We are also the first in the literature to coherently describe and explain how future trials should be conducted in this field to optimise the care burn patients receive; to ensure patients in both high and low-middle income countries receive the most effective burn wound dressing. Finally, we also comment on the lack of consistent outcome measures (specifically secondary outcomes) between studies and give detail of suitable outcomes all future trials in this field should report.

In addition to the above comments, all spelling and grammatical errors pointed out by the reviewers have been corrected. We look forward to hearing from you in due time regarding our submission and to respond to any further questions and comments you may have.

Sincerely,

Samatar Osman

5th year Intercalating Medical Student (MSc Trauma Science)

University of Birmingham

Round 2

Reviewer 2 Report

The originality rate of this paper is still low. This study is acceptable as it is a review article and most of the proposed revisions have been made.